# BiEnhancer: Bi-Level Feature Enhancement in the Dark

## Abstract

The remarkable achievements of high-level vision tasks (e.g., object detection, semantic segmentation) under favorable lighting conditions highlight the persistent challenges faced in low-light vision. Previous studies have mainly focused on enhancing low-light images to create visual-friendly representations, often neglecting the differences between machine vision and human vision. This oversight has led to limited performance improvements for high-level tasks. Furthermore, many approaches rely on synthetic paired datasets for training, which can result in limited generalization to real-world images with diverse illumination levels. To address these issues, we propose a new module called BiEnhancer, which is designed to enhance the representation of low-light images by optimizing the loss function of high-level tasks to improve performance. BiEnhancer decomposes low-light images into low-level and high-level components and performs feature enhancement. Then, it adopts an attentional feature fusion strategy and a pixel-wise iterative estimation strategy to effectively enhance and restore the details and semantic information of low-light images and improve the machine-readable representation ability of low-light images. As a versatile plug-in module, BiEnhancer supports end-to-end joint training with diverse high-level tasks. Extensive experimental results demonstrate that the BiEnhancer framework outperforms state-of-the-art methods in both speed and accuracy.

## 1 Introduction

Computer vision has achieved remarkable success in processing high-quality images and videos. Existing backbone networks (Gao et al., 2019) (Dosovitskiy, 2020) (Liu et al., 2021b) (Chen et al., 2023), object detectors (Ren et al., 2016) (Redmon, 2016) (Redmon, 2018) (Meng et al., 2021) (Zhang et al., 2022) (Wang et al., 2024) (Zhao et al., 2024), and semantic segmentation models (Long et al., 2015) (Qin et al., 2020) perform well on benchmark datasets. However, although existing low-light image enhancement (LIE) methods (Zhang et al., 2019) (Wang et al., 2023) (Jin et al., 2023) have shown significant improvements in converting low-light images into visual-friendly representations, high-level vision tasks such as object detection and semantic segmentation still face challenges under low-light conditions. Simply combining LIE models and high-level vision tasks may not necessarily lead to improved performance.

There are two methods for combining LIE models with high-level vision tasks: end-to-end training and Non end-to-end training. End-to-end training entails removing the loss function of the LIE model (Hashmi et al., 2023). Low-light images processed by the LIE model are fed into the high-level framework's backbone network, and both models optimize synchronously using the high-level model's loss function. Non end-to-end training first trains the LIE model on paired data with its own loss function, then sends enhanced images to the high-level model for fine-tuning. The comparison in Figure 1 reveals the inferior performance of simple model combinations (Non end-to-end training) in terms of both detection accuracy and efficiency.

This work explores the underlying reasons for the low performance observed in a Non end-to-end training of LIE methods with high-level vision tasks: 1) The different loss functions employed by LIE and high-level vision models lead to optimization conflicts between them, resulting in the performance of the entire combination being less than expected. End-to-End training can better coordinate the two parts and achieve global optimality. For example, the loss functions of Sparse

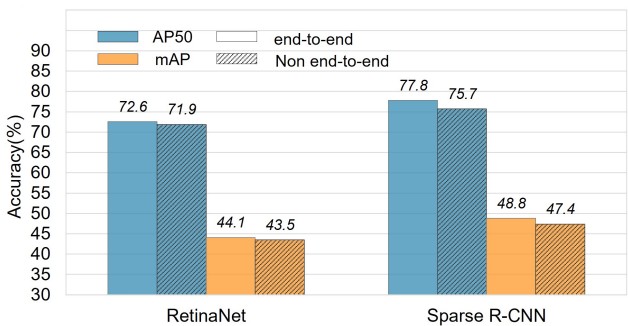

Figure 1: Performance comparison of Zero-DCE when trained end-to-end or Non end-to-end with RetinaNet and Spark R-CNN as benchmarks on the ExDark dataset.

R-CNN (Sun et al., 2021), including L1 Loss, GIoU Loss (Rezatofighi et al., 2019), and Focal Loss, prioritize improving prediction accuracy. On the other hand, the loss functions of Zero-DCE (Guo et al., 2020), including Spatial Consistency Loss, Exposure Control Loss, Color Constancy Loss, and Illumination Smoothness Loss, focus on enhancing image brightness and contrast to improve image clarity under low-light conditions. However, this enhancement may compromise the semantic information crucial for accurate detection. 2) Current LIE methods (Liu et al., 2021a) (Ma et al., 2022) often emphasize low-level features at the expense of enhancing high-level features rich in semantic information (see in Figure 4. In the ground truth (GT), there are four boats. However, Zero-DCE only identifies two of them. On the contrary, there is no person in GT, but Zero-DCE misidentifies some objects as a person). Although restoring low-level features can improve local details, due to the lack of basic semantic context essential for accurately understanding images, restoring low-level features may ultimately reduce the machine readability of the entire image.

To address two issues, we propose BiEnhancer, a multifunctional plug-in module. Unlike traditional LIE methods (Guo & Hu, 2023) (Cui et al., 2021), , it doesn't rely on a specific loss function and can train end-to-end with high-level vision task models. BiEnhancer uses a feature aggregation enhancement strategy and an attentional bi-level feature fusion strategy inspired by cross attention (Vaswani, 2017). It also employs a pixel-wise iterative estimation strategy. These strategies enhance robustness and improve performance and efficiency of tasks. Experimental results show BiEnhancer outperforms existing state-of-the-art methods and improves results in low-light vision tasks, e.g., +0.5 mAP and +0.5 FPS in dark object detection on ExDark, +0.9 mAP in face detection on DARK FACE, and +0.3 mIoU and +0.5 FPS in nighttime semantic segmentation on ACDC with an A5000 GPU.

Our main contributions can be summarized as follows:

(*i*) We propose BiEnhancer, a novel module that can improve the extraction of low-level features and enhance the semantic quality of high-level features to boost high-level vision tasks under low-light conditions

(*ii*) We introduce the attentional bi-level feature fusion strategy to effectively fusing low-level and high-level features.

(*iii*) Our proposed pixel-wise iterative estimation strategy can rapidly iterate low-light images to obtain more machine-readable and feature-enhanced representations.

## 2    RELATED WORK

### 2.1    LOW-LIGHT ENHANCEMENT

Most LIE methods (Lore et al., 2017) (Moran et al., 2020) are designed to transform low-light images into visual-friendly representation by increasing brightness, restoring color, completing details, reducing noise, etc. Traditional LIE methods (Ibrahim & Kong, 2007) (Lee et al., 2013) mainly rely on histogram equalization (Pizer et al., 1987) and Retinex theory (Land & McCann, 1971), but

often face challenges such as increased noise, color distortion, and slow processing speed. Recently, deep learning has made significant progress in LIE, and LIE methods can be categorized into supervised learning, unsupervised learning, semi-supervised learning, and zero-reference learning according to different learning strategies (Li et al., 2021). Supervised learning is mainstream, although it performs well, it often lacks generalization ability in real low-light conditions due to the use of synthetic training data. To address these issues, unsupervised EnlightenGAN (Jiang et al., 2021) applied GAN (Goodfellow et al., 2014) technology for the first time in low-light fields. This approach enables training using unpaired data while employing discriminators to handle different lighting conditions, albeit with a somewhat unstable training process. To combine the advantages of supervised and unsupervised learning, semi-supervised DRBN (Yang et al., 2020) is proposed, which performs unsupervised band reassembly after supervised recursive band learning, but it is difficult to construct cross-domain information relationships. To compensate for these shortcomings, Guo et al. (2020) propose zero-reference learning, which only enhances learning from test images, but faces the challenge of designing non-reference loss functions.

### 2.2 Low-Light Enhancement for Downstream Visual Tasks

Research on object detection under sub-optimal lighting conditions (including low light conditions) has resulted in several YOLOv3-based methods (Liu et al., 2022) (Kalwar et al., 2023) to improve detection performance. However, these methods often struggle with adaptability to other models. Cui et al. (2021) introduced MAET to improve object detection performance under low-light conditions by analyzing intrinsic lighting patterns. Their subsequent work, IAT (Cui et al., 2022), employed the Transformer architecture to estimate global image signal processing (ISP) parameters for improving object detection and semantic segmentation performance. However, this might come at the cost of slower detection speeds. Based on the contrastive learning strategy, Xue et al. (2022) designed a joint unified framework with a cascaded architecture that can enhance the visual and machine perception capabilities of nighttime semantic segmentation. Ma et al. (2022) established a cascaded illumination learning process (SCI) with weight sharing to enhance the visual quality of low light images and improve performance in tasks such as low-light face detection and nighttime semantic segmentation. Hashmi et al. (2023) proposed a FeatEnHancer network with no loss function, which aggregates and generates multi-level features to enhance the performance of advanced visual tasks. Our work also adheres to this spirit, and the proposed BiEnhancer performs better and faster in advanced visual tasks.

## 3 Methods

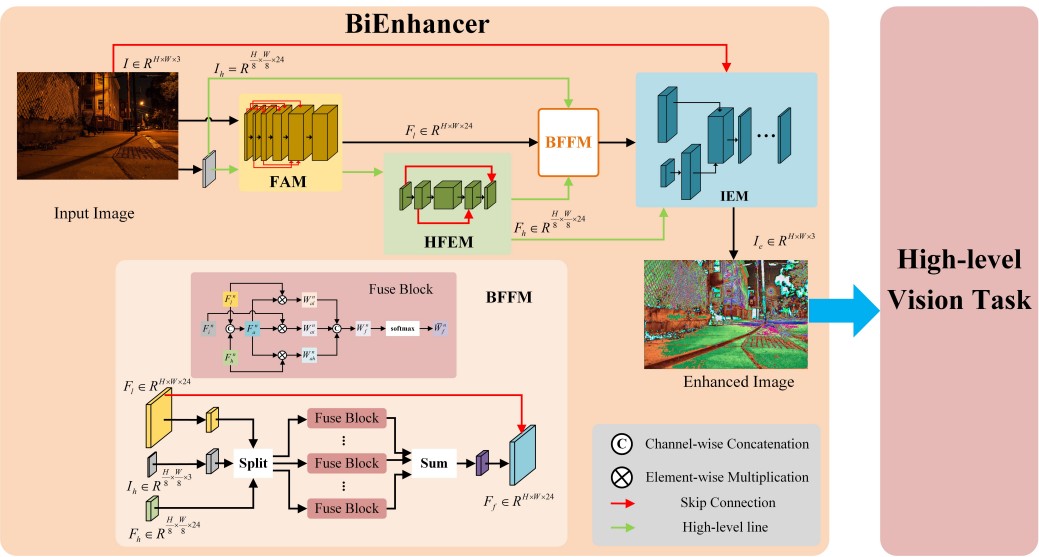

Figure 2: The overview of the framework of BiEnhancer.

## 3.1 OVERVIEW OF BIENHANCER

This paper introduces BiEnhancer, a versatile and robust plug-in module specifically designed to enhance machine readability under low-light conditions for vision tasks such as object detection, face detection, and semantic segmentation. BiEnhancer decouples low-light images into low-level and high-level features, enhances them, and effectively fuses them while preserving details and strengthening their semantic description. Finally, the enhanced low-light representation is obtained through pixel-level iterative evaluation. An overview of the framework of BiEnhancer is presented in Figure 2.

## 3.2 FEATURE EXTRACTION AND ENHANCEMENT

In this section, we take a low-light RGB image $I \in \mathbb{R}^{W \times H \times 3}$ as input. An reflection convolutional operator **RefConv** (an regular convolutional operator with reflection padding) is employed on $I$ to generate a low-resolution image $I_l \in \mathbb{R}^{\frac{H}{8} \times \frac{W}{8} \times 3}$. Then, the Feature Aggregation Module (FAM) is utilized to transform $I$ and $I_l$ into low-level features $F_l \in \mathbb{R}^{W \times H \times C}$ and high-level features $f_h \in \mathbb{R}^{\frac{H}{8} \times \frac{W}{8} \times C}$. Subsequently, the High-level Feature Enhancement Module (HFEM) processes to generate richer high-level features $F_f \in \mathbb{R}^{\frac{H}{8} \times \frac{W}{8} \times C}$.

**Feature Extraction** Inspired by DenseNet (Huang et al., 2017), we design a fully convolutional inter-scale Feature Aggregation Module (FAM) for aggregating features and capturing crucial spatial and channel information. We apply four convolution blocks to the RGB image $I$ to generate the aggregated high-level features $F_h \in \mathbb{R}^{W \times H \times C}$. In each convolution block, the input is the sum obtained by concatenating the output of the previous block and the original image $I$. Each convolution block is accompanied by a SiLU activation function. In fact, BiEhnancer only uses the SiLU activation function. The process of obtaining $f_h$ by processing $I_h$ with FAM is similar to the above, and the structure of FAM is shown in Figure 3(a).

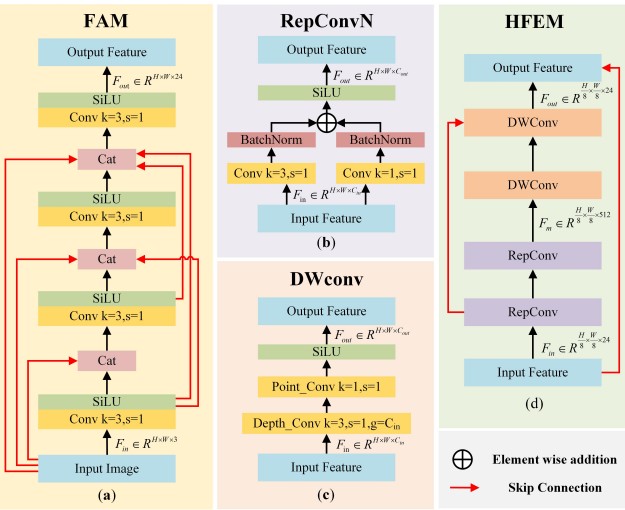

Figure 3: Details of FAM, ReponvN, DWconv and HFEM.

**Feature Enhancement** High-level features embody rich semantic detail and serve as crucial elements for downstream visual tasks. To extract this semantic information, we were motivated by U-net (Ronneberger et al., 2015) and designed the high-level feature enhancement module (HFEM). In HFEM, we utilize depthwise separable convolution (DW Conv) (Chollet, 2017) and simplified reconparameterization convolution (RepConvN) (Ding et al., 2021) bolocks as the basic operation unit. The structures of the RepConvN and DWConv blocks are shown in Figure 3(b) and 3(c). We start with two RepConvN blocks to convert low-channel features $f_h$ into a high-channel features, followed by two DWConv blocks reduce the channels and obtain the final high-level feature $F_h \in \mathbb{R}^{\frac{H}{8} \times \frac{W}{8} \times C}$. In HFEM, in order to obtain more comprehensive semantic information, we also used two skip connections (SC). The structure of HFEM is shown in Figure 3(d).

## 3.3 FEATURE FUSION

Single low-level or high-level features are difficult to effectively improve vision task performance, while fusion features can better reflect the complexity of visual information (as shown in Tables 5). In Bi-level Feature Fusion Module (BFFM), we propose an efficient feature alignment and attentional fusion scheme to enhance the ability of module to represent low-level details and high-level semantics. As shown in Figure 2, to maintain the lightweight BiEnhancer, we perform feature alignment at the low-resolution scale by using a reflection convolutional operator **RefConv** (K=9, S=8) to down-sampling the low-level features $F_l \in \mathbb{R}^{H \times W \times C}$ and another regular convolutional operator **Conv** (K=3, S=1) to increase the number of channels for $I_h$. Then, we split the down-sampled low-level features $F_l \in \mathbb{R}^{\frac{H}{8} \times \frac{W}{8} \times C}$, $F_i \in \mathbb{R}^{\frac{H}{8} \times \frac{W}{8} \times C}$ and the high-level features $F_h \in \mathbb{R}^{\frac{H}{8} \times \frac{W}{8} \times C}$ into $N$ blocks along the channel dimension C. It can be written as:

$$F_l^n = F_l[:,:,(n-1)\frac{C}{N}:n\frac{C}{N}]; F_i^n = F_i[:,:,(n-1)\frac{C}{N}:n\frac{C}{N}]; F_h^n = F_h[:,:,(n-1)\frac{C}{N}:n\frac{C}{N}] \tag{1}$$

where $n \in 1,2,...,N$ and $N$ is the number of fusion blocks. In a single fusion block, $F_i^n \in \mathbb{R}^{H \times W \times 1 \times 1 \times \frac{C}{N}}$ and $F_h^n \in \mathbb{R}^{H \times W \times 1 \times 1 \times \frac{C}{N}}$ are concatenated along the third dimension L to obtain $F_a^n \in \mathbb{R}^{H \times W \times 2 \times 1 \times \frac{C}{N}}$. The three features $F_l^n \in \mathbb{R}^{H \times W \times 1 \times 1 \times \frac{C}{N}}$, $F_i^n$ and $F_h^n$ are multiplied with $F_a^n$ separated and then summed up along the last dimension T to obtain the attentional weights $W_{al}^n \in \mathbb{R}^{H \times W \times 2 \times 1 \times 1}$, $W_{ai}^n \in \mathbb{R}^{H \times W \times 2 \times 1 \times 1}$ and $W_{ah}^n \in \mathbb{R}^{H \times W \times 2 \times 1 \times 1}$. It can be written as:

$$W_{al}^n = \sum_{t=-1}^{T} (F_a^n \cdot F_l^n \cdot s); \quad W_{ai}^n = \sum_{t=-1}^{T} (F_a^n \cdot F_i^n \cdot s); \quad W_{ah}^n = \sum_{t=-1}^{T} (F_a^n \cdot F_h^n \cdot s) \tag{2}$$

where $\sum_{t=1}^{T}$, $\cdot$ ,and $s$ denote the summing operation along the last dimension T, the element-wise multiplication operation, and the scale factors ($N^{-0.5}$). Then, we concatenate $W_{al}^n$, $W_{a0}^n$ and $W_{ah}^n$ along the last dimension to obtain the total attentional weights of a fuse block $W_f^n \in R^{H \times W \times 2 \times 1 \times \frac{C}{N}}$. Due to the limitation of the above concatenation calculation, it is necessary to ensure that $\frac{C}{N}$ is equals to 3. Therefore, in this paper, we set $C$ to 24 and $N$ to 8. The process can be written as:

$$W_f^n = Cat([W_{al}^n, W_{a0}^n, W_{ah}^n], dim = T); \quad \overline{W_f^n} = \frac{\exp(W_f^n)}{\sum_{t=1}^{T} \exp(W_f^n)} \tag{3}$$

where Cat([·],dim=T) represents the concatenation operation along the last dimension T, and $\overline{W_f^n}$ is the normalized form of $W_f^n$. Then, we sum up all $\overline{W_f^n}$ along the dimension L to obtain $\overline{W_f} \in \mathbb{R}^{\frac{H}{8} \times \frac{W}{8} \times C}$. Finally, we use the bilinear interpolation operator **BiUp** to upsample $\overline{W_f}$ to obtain $W_f \in \mathbb{R}^{H \times W \times C}$, and then add the original low-level features $F_l \in \mathbb{R}^{W \times H \times C}$ to get the fused feature representation $F_f \in \mathbb{R}^{H \times W \times C}$.

## 3.4 ITERATIVE ESTIMATION

Inspired by the denoising process of DDPM (Ho et al., 2020), we design the Iterative Estimation Module (IEM) to further optimize and enhance the feature representation of low-light images. First, bilinear interpolation operator **BiUp** is utilized to up-sample $F_h$ by a factor of 8 to attain the original resolution size. Subsequently, the up-sampled $F_h$ is added to the fused feature representation $F_f$ in the channel dimension. Next, reflection convolution operator **RefConv** is employed to operate on the added feature representations $F_a \in \mathbb{R}^{H \times W \times 2C}$, obtaining all parameters for iterative evaluation. We set $F_a$ to 2*N (twice the number of fusion blocks) parameters $\bar{F}_n$ that are of the same size as the low-light image $I$. Then, $\bar{F}_n$ is used to iteratively evaluate $I$ and transform it into a feature-enhanced

representation $F_e$ that is more machine-readable for high-level vision tasks. The iterative estimation process can be written as:

$$I_{n+1} = I_n \left(1 + \bar{F}_{2n-1} + \bar{F}_{2n} I_n\right) \tag{4}$$

Here, $n \in 1, 2, ..., N$, and N is the number of iterations. $I_1$ is the low-light RGB image $I$, and $I_{N+1}$ is the feature enhancement representation $F_e$.

# 4 EXPERIMENTS

We evaluate the effectiveness of BiEnhancer through extensive experiments on several high-level tasks in low-light vision, including generic object detection, face detection, and semantic segmentation. This section compares the proposed method with powerful baselines, existing LIE methods, and state-of-the-art methods in these high-level tasks. We conducted ablation experiments on different BiEnhancer blocks to assess their effectiveness. The key statistical data of datasets is summarized in Table 1.

| Datasets | Task | Cls | Train | Val |
|---|---|---|---|---|
| ExDark | Object detection | 12 | 4800 | 2563 |
| DARK FACE | Face detection | 1 | 5400 | 600 |
| ACDC Nightime | Semantic segmentation | 19 | 400 | 106 |

Table 1: Statistics of the datasets used to report results on three different downstream vision tasks. **Cls** is the number of classes, whereas **Train** and **Val** denote number of training and validation samples for each dataset, respectively.

## 4.1 DARK OBJECT DETECTION ON EXDARK

**Details** For dark object detection, we used the exclusively dark (ExDark) [1] dataset (see in Table 1) in our experiments using RetinaNet (Lin, 2017), a typical detector, and Sparse R-CNN, an advanced detector as detection frameworks. Both detectors were initialized with pre-trained weights from COCO dataset and fine-tuned on the ExDark dataset using multiscale training (shorter sides 320 to 520, longer side 608). RetinaNet was trained with a 1×schedule in mmdetection [2] (12 epochs using the SGD optimizer with an initial learning rate of 0.01, and batch size of 8). The Sparse R-CNN was trained with a 1×schedule in mmdetection (12 epochs using the ADAMW optimizer (Loshchilov et al., 2017) with an initial learning rate initial learning rate of 0.000025, weight decay of 0.0001, and batch size of 8).

We compared our BiEnhancer with several leading LIE methods, including SCI, MBLLEN (Lv et al., 2018), RAUS, PairLIE (Fu et al., 2023), Retinexformer (Cai et al., 2023), Zero-DCE, Zero-DCE++, and state-of-the-art dark object detection method, FeatEnHancer. For each object detection framework, we maintained the identical settings and employed the same end-to-end joint training approach, where the low-light image is propagated to the detector after passing through the enhancement network, without any LIE loss function.

**Results** Table 2 lists the results of LIE methods, FeatEnHancer, and our proposed BiEnhancer on two object detection frameworks. Clearly, our BiEnhancer consistently offers enhanced detection precision and faster test speeds, outperforming previous approaches. Specifically, on the Sparse R-CNN framework, our BiEnhancer outperforms FeatEnHancer's AP50 by 1.1%, reaching a score of 78.9, and its mAP is also superior by 0.5%. Furthermore, on the RetinaNet framework, our BiEnhancer shows more efficacy than FeatEnHancer (+0.2 AP50 and +0.6 mAP). Concurrently, when using an A5000 GPU on both detection platforms, BiEnhancer proves swifter than all current advanced LIE methods, achieving a speed advantage of 0.5 FPS over FeatEnHancer on Sparse R-CNN. In addition, Figure 4 shows four detection examples from our method and two best competitors using Sparse R-CNN as the detector. These results indicate that despite poor visual quality, our BiEnhancer enhances and integrates beneficial features for detecting dark objects, producing industry-leading results.

---

[1] https://github.com/cs-chan/Exclusively-Dark-Image-Dataset
[2] https://github.com/open-mmlab/mmdetection

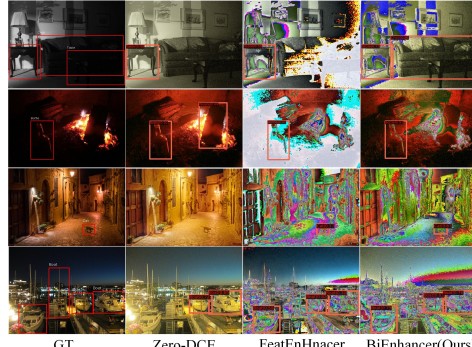

Figure 4: Visual comparison of Bienhancer with SCI, Zero-DCE, and FeatEnHancer on the Dark Face dataset.

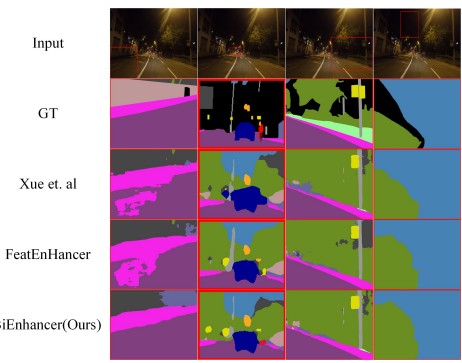

Figure 5: Visual comparison of Bienhancer with the two previous best competitors on the ACDC nighttime dataset

| Methods | Sparse R-CNN | | | RetinaNet | |
|---|---|---|---|---|---|
| | AP50 | mAP | FPS | AP50 | mAP |
| Baseline | 76.3 | 47.8 | 46.8 | 71.0 | 42.5 |
| RUAS | 67.3 | 40.9 | 38.4 | 73.3 | 44.7 |
| Retinexformer | 73.6 | 45.6 | 14.3 | 73.5 | 44.5 |
| MBLLEN* | 76.4 | 47.4 | 16.5 | 71.2 | 42.5 |
| SCI | 76.5 | 48.1 | 31.3 | 72.5 | 44.0 |
| PairLIE* | 77.0 | 48.2 | 26.7 | 71.8 | 43.5 |
| Zero-DCE++ | 77.4 | 48.6 | 38.5 | 73.4 | 44.5 |
| Zero-DCE | 77.8 | 48.8 | 39.6 | 72.6 | 44.1 |
| FeatEnHancer | 77.8 | 48.8 | 36.4 | 73.6 | 44.6 |
| BiEnhancer (Ours) | 78.9 | 49.3 | 40.1 | 73.8 | 45.2 |

Table 2: Performance comparison on Exdark dataset. The best and second-best results are marked in red and blue respectively. Note that here MBLLEN* and PairLIE* denote training with a skip connection (SC).

## 4.2 FACE DETECTION ON DARK FACE

**Details** DARK FACE is a challenging face detection dataset released for the UG2 competition. For dark object detection, we utilized the DARK FACE dataset [3] (see Table 1) in our experiments with RetinaNet and Sparse R-CNN detection frameworks. For experiments on DARK FACE (see in Table 3), images are resized to a resolution of 1080×720 for both methods. To emulate our previous findings on Dark Object Detection experiments, we utilize the identical RetinaNet and Sparse R-CNN objects detection frameworks, maintaining the same prescribed experimental conditions. For Sparse R-CNN, the batch size has been adjusted to to 4.

**Results** The performance of BiEnhancer, FeatEnHancer, and seven other LIE methods in combination with RetinaNet and Sparse R-CNN is summarized in Table 3. Similarly, our BiEnhancer has significantly improved the detection accuracy and speed on both Sparse R-CNN and RetinaNet compared to FeatEnHancer. On Sparse R-CNN, both AP50 (+1.2) and mAP (+0.9) are higher than those of FeatEnHancer. On RetinaNet, AP50 (+1.4) and mAP (+0.3) are also higher than those of FeatEnHancer.

---

[3] https://flyywh.github.io/CVPRW2019LowLight/

| Methods | Sparse R-CNN | | RetinaNet | |
|---|---|---|---|---|
| | AP50 | mAP | AP50 | mAP |
| Baseline | 52.5 | 21.6 | 32.6 | 12.3 |
| RUAS | 51.5 | 21.1 | 36.8 | 14.1 |
| MBLLEN* | 54.2 | 22.3 | 32.4 | 12.4 |
| SCI | 54.3 | 22.4 | 37.0 | 14.1 |
| Retinexformer | 56.6 | 22.9 | 36.8 | 13.9 |
| PairLIE* | 56.3 | 23.3 | 34.6 | 13.0 |
| Zero-DCE++ | 57.7 | 23.9 | 36.3 | 13.9 |
| Zero-DCE | 58.8 | 24.4 | 37.1 | 14.0 |
| FeatEnHancer | 58.5 | 24.4 | 36.4 | 13.8 |
| BiEnhancer (Ours) | 60.0 | 25.5 | 38.5 | 14.4 |

Table 3: Performance comparison on the Dark Face dataset. The best and second-best results are marked in red and blue respectively. Note that here MBLLEN* and PairLIE* denote training with a skip connection (SC).

## 4.3 NIGHTTIME SEMANTIC SEGMENTATION ON ACDC

**Details** We utilize nighttime images from the ACDC dataset [4] (see in Table 1) to report semantic segmentation results under low-light conditions. DeepLab-V3 (Chen, 2017) and SegFormer (Xie et al., 2021) was adopted as the segmentation baseline from mmsegmentation [5] for straightforward comparison with concurrent works. The module is initialized with pretrained weights from Cityscapes and fine-tuned on the ACDC nighttime dataset, while images are resized to a resolution of 1920×1080. DeepLab-V3 is trained with a 40k schedule in mmsegmentation (40000 iterations using the SGD optimizer, an initial learning rate of 0.001, weight decay of 0.0005, and a batch size of 4), and the crop size is set to (512, 1024). SegFormer is trained with a 20k schedule in mmsegmentation (20000 iterations using the ADAMW optimizer with an initial learning rate of 0.00005, weight decay of 0.01, and a batch size of 8), and the crop size is set to (1024, 1024).

**Results** Table 4 summarizes the performance of BiEnhancer, FeatEnHancer, and seven other LIE methods in combination with SegFormer and DeepLab-V3. Our BiEnhancer has brought significant baseline improvements. On DeepLab-V3, the mIoU is 53.1, which is 0.3 higher than the previous best result. On SegFormer, the mIoU is 54.4, also 0.2 higher than the previous best result. In addition, our BiEnhancer shows a segmentation speed of 4.5 FPS on the A5000 GPU while maintaining excellent segmentation accuracy. It is significantly improved by 0.5 FPS compared to FeatEnHancer. At present, we have made a qualitative comparison with the previous best competitor in Figure 5. Obviously, our BiEnhancer can generate more accurate segmentation for both larger and smaller objects. These results confirm the effectiveness of BiEnhancer as a general-purpose module to achieve state-of-the-art results in nighttime semantic segmentation.

## 4.4 ABLATION STUDIES

In this section, we conduct ablation studies on the key design components of the proposed BiEnhancer when integrated with Sparse R-CNN for dark object detection on the ExDark dataset and DeepLab-V3 for nighttime semantic segmentation on the ACDC dataset. As shown in Table 5, we emphasize the evaluation results of dark object detection and nighttime semantic segmentation obtained by removing individual critical elements within the BiEnhancer framework.

**Scale of High-level Features** As shown in Table 6, varying the configuration of the inaugural two halved convolutional modular units on Image I resulted in different resolution levels for high-level features $F_h$. Despite a marginal increase in object detection speed (+0.2 FPS), it's noteworthy that 8X down-sampling provides optimal results when considering BiEnhancer as a whole.

---

[4] https://acdc.vision.ee.ethz.ch/download
[5] https://github.com/open-mmlab/mmsegmentation

| Methods | DeepLab-V3 | | SegFormer |
| | mIoU | FPS | mIoU |
| --- | --- | --- | --- |
| Baseline | 50.0 | 5.2 | 52.8 |
| RUAS | 49.8 | 4.3 | 52.5 |
| SCI | 50.8 | 3.4 | 52.9 |
| Xue et al. | 52.3 | 4.6 | 52.9 |
| PairLIE* | 52.5 | 2.9 | 53.8 |
| MBLLEN* | 52.6 | 2.4 | 53.3 |
| Zero-DCE++ | 52.6 | 4.3 | 53.5 |
| Zero-DCE | 52.7 | 4.3 | 54.1 |
| FeatEnHancer | 52.8 | 4.0 | 54.2 |
| **FFNet(Ours)** | 53.1 | 4.5 | 54.4 |

Table 4: Performance comparison on ACDC nighttime dataset. The best and second-best results are marked in red and blue respectively. Note that here MBLLEN* and PairLIE* denote training with a skip connection (SC).

| FAM | HFEM | BFFM | IEM | Exdark mAP | ACDC mIoU |
| --- | --- | --- | --- | --- | --- |
| | | | | 47.8 | 50.0 |
| | ✓ | ✓ | ✓ | 48.0 | 50.7 |
| ✓ | | ✓ | ✓ | 48.7 | 52.3 |
| ✓ | ✓ | | ✓ | 48.6 | 51.8 |
| ✓ | ✓ | ✓ | | 48.5 | 51.4 |
| ✓ | ✓ | ✓ | ✓ | **49.3** | **53.1** |

Table 5: Effectiveness of BiEnhancer.

| Scale | ExDark | | ACDC | |
| | mAP | FPS | mIoU | FPS |
| --- | --- | --- | --- | --- |
| 2 | 48.6 | 30.5 | 52.3 | 3.5 |
| 4 | 48.9 | 37.5 | 52.5 | 4.1 |
| 8 | **49.3** | 40.1 | **53.1** | 4.5 |
| 16 | 48.6 | 30.5 | **52.2** | **4.7** |

Table 6: Various combinations of the scale of high-level features. Here, 4 means the high-level features is $F_h \in \mathbb{R}^{\frac{H}{4} \times \frac{W}{4} \times C}$.

| N | C | ExDark | | ACDC | |
| | | mAP | FPS | mIoU | FPS |
| --- | --- | --- | --- | --- | --- |
| 2 | 6 | 48.0 | **44.2** | 50.3 | **4.8** |
| 4 | 12 | 48.4 | 37.5 | 52.5 | 4.7 |
| 8 | 24 | **49.3** | 40.1 | **53.1** | 4.5 |
| 16 | 48 | 48.9 | 30.5 | 52.9 | 4.2 |

Table 7: Various combinations of the number of fusion blocks.

**Number of Fusion Blocks in BFFM** As shown in Table 7, variable N (number of fusion blocks in the proposed BFFM model) has a significant impact. Increasing N improves set accuracy and test speed, but when N exceeds 8, object detection and semantic segmentation speeds slow down. Marginal accuracy improvement doesn't justify the performance degradation. Optimal number of fusion blocks is 8.

## 5 CONCLUSION

This paper presents a new multifunctional plug-in module called BiEnhancer, which is designed to enhance the fused bi-level features crucial for low-light vision tasks. Our feature aggregation and enhancement scheme is aligned with the vision backbone network, producing robust semantic representations. BiEnhancer does not require pre-training on synthetic datasets nor rely on enhancement loss functions, making it a plug-and-play solution. Extensive experiments across three different vision tasks demonstrate that our method consistently outperforms baseline models, LIE models, and leading approaches for specific tasks.

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
