# OpenReview forum: "BiEnhancer: Bi-Level Feature Enhancement in the Dark"
_ICLR.cc/2025/Conference — Submitted to ICLR 2025_

### Official Review · Reviewer_AsJr · 2024-10-27

**Soundness:** 3
**Presentation:** 2
**Contribution:** 3
**Rating:** 5
**Confidence:** 3

**Summary:**

This paper studies the problem of low-light image enhancement to support high-level vision tasks. Its key idea is to design a plug-and-play module called BiEnhancer that consists of feature extraction, feature fusion, and iterative estimation. The proposed approach was shown to outperform FeatEnHancer (ICCV'2023) the previous SOTA benchmark. Performance improvements (based on Tables 2-4) are modest - .e.g., 0.3-1.1 in terms of mAP and mIoU.

**Strengths:**

+ Originality: The idea presented in Sec. 3.4 (Iterative estimation) is novel and interesting. It borrows the idea from diffusion model to iteratively optimize the feature representation of low-light images.
+Significance: The reported experimental results have shown consistent improvement over all benchmark methods including Zero-DCE and FeaEnHancer. In some situation, the performance gain can be over 1 point (e.g., Table 3).

**Weaknesses:**

-Quality: I have found the literary presentation of this work significantly lacking. The main body of this work (Sec. 3) is difficult to read, especially Sec. 3.2 and 3.3. Those sections contain lots of technical details (how we do it?) without discussing motivation or new insight. The interesting idea inspired by DDPM was only briefly mentioned in Sec. 3.4, which makes it difficult to appreciate its elegance.
-Clarity: I have been struggling all the way with identifying the authors' contribution from the standard practice in the literature. For example, Fig. 3 shows the details of three architectures but did not say anything about their relationship, which one or which part represents the authors' new design. The feature fusion formula (i.e., Eq. (3) in Sec. 3.3) is also presented without any substantial justification.
-Technical contribution: Fig. 2, to me, is still an open-loop design. Such open-loop design has fundamental limitations because it lacks the feedback from high-level vision tasks. A more fruitful approach, in my biased opinion, is a closed-loop design that jointly optimize low-level and high-level vision tasks. Note that in an open-loop design, any errors made by BiEnhancer can not be corrected  later - e.g., the amplified noise as shown in Fig. 2 has been totally overlooked by the authors.
-Experimental results. I took a look at the mAP results reported by FeatEnHancer (https://github.com/khurramHashmi/FeatEnHancer). On the ExDark and Dark Face datasets, they are very different from those reported in this paper. I suspect that there are some parameter setting issues. But the missing of reference for FeatEnHancer (Line 307) seems unforgivable because this is the most important reference in benchmark methods.

**Questions:**

1) Have you investigated the issue of noise amplification in your study? From Fig. 2, one cannot help wondering if the amplified noise, without proper guidance, might have a negative impact on the high-level vision tasks.
2) What could cause the result inconsistency between ICCV'2023 and your experiments? In FeatEnHancer + Featurized Query R-CNN, mAP of 86.3 and 69.0 were reported for ExDark and Dark Face datasets. Why did you have 49.3 and 25.5 (according to Tables 2 and 3)?
3) Can you elaborate on the design of iterative estimation in Eq. (4)? I think this equation might contain some error because the terms within the parenthesis appear inconsistent (i.e., the last term contains I_n).
4) What is the rationale for visual comparison in Fig. 4? It seems to me that visual quality is not the purpose of BiEnhancer, right? Visually, we should not expect any meaningful results from enhanced images.

---

### Official Review · Reviewer_A3cw · 2024-10-28

**Soundness:** 2
**Presentation:** 2
**Contribution:** 2
**Rating:** 3
**Confidence:** 5

**Summary:**

This paper proposes to enhance the performance of low-light image enhancement for downstream high-level tasks. The proposed method is called BiEnhancer, which optimize the loss function of high-level tasks to improve performance. Moreover,  BiEnhancer employs an attentional feature fusion and a pixelwise iterative estimation strategy to effectively restore the details which can help the semantic information.

**Strengths:**

This paper optimizes the high-level tasks' loss besides the low-light enhancement target, and the improvement on the downstream high-level tasks is achieved.

**Weaknesses:**

1. Although this paper is designed as a bi-level framework, this paper mainly consider the improvement on the high-level tasks, and ignore the visual improvement for the human vision. As we can see in the Fig.2 and Fig.4, the enhanced image is difficult to see without suitable color and illuminations. The bi-level target should be enhance the low-light image which is both human- and machine-friendly.

2. Although this framework has sacrificed the visual quality, the improvement on the downstream tasks is not very obvious, and improvements are almost 1-2 points.

3. There is not sufficient novelty or novel motivation for the network design, e.g., the skip connections, RepConv, and DWConv are all common network architectures in the computer vision field.

4. The presentation of this paper can be further improved. For example, the tables of 3 and 4 are of over-large size, while Figures 4 and 5 are not large enough for visualization.

**Questions:**

1. What is the visual quality for the enhanced images with BiEnhancer? Show the quantitative results.

2. The comparison with more SOTA low-light image enhancement methods should be provided.

---

### Official Review · Reviewer_fiJA · 2024-10-29

**Soundness:** 2
**Presentation:** 2
**Contribution:** 1
**Rating:** 3
**Confidence:** 5

**Summary:**

This manuscript proposed a bi-level feature enhancer to boost the performance of low-light object detection. Experiments show the superiority over previous feature enhancement methods for both performance and processing speed.

**Strengths:**

+ The proposed method performs better than the previous method, FeatEnhancer.

**Weaknesses:**

+ Motivation. BiEnhancer proposed a "plug-and-play" method for low-light images. However, the proposed BiEnhancer still requires training on the specific detector and low-light image dataset. It can't be applied to other detectors/normal-light images without fine-tuning. However, the low-light enhancers can be applied to various pre-trained detectors. Besides, I think the concept of "feature enhancement" makes no sense. If you assign more parameters and multi-scale mechanisms to the backbone, the performance will gain, but one will never call it a **low-light enhancer**.  The community is not **"neglecting"** the difference between machine vision and human vision, it is just off-topic for enhancement methods to feed the detector up.
+ Weak baselines.  As time flies, numerous stronger methods have been proposed in the literature, some of them are even cited by the authors in the related work part of the manuscript. I believe some of them can perform better than these 2 baselines.
+ More modern detectors. The proposed method only examines RetinaNet and Sparse R-CNN. However, more modern detection frameworks typically involve stronger augmentation techniques and better pretrain (Object365, for instance). I wonder if the proposed method still works under these new settings.
+ Novelty. What is the difference between the proposed method and FeatEnhancer? This part should be highlighted.
+ Unfair comparison. It seems that the parameters of image enhancers are fixed while the Bi-Enhancer can learn from high-level loss. This is unfair, I wonder if you can unlock the parameters of these enhancers to see what happens.

**Questions:**

Please, refer to the weakness part where I have stated my concerns about this work.

---

### Official Review · Reviewer_twtu · 2024-10-30

**Soundness:** 2
**Presentation:** 2
**Contribution:** 2
**Rating:** 3
**Confidence:** 5

**Summary:**

The paper proposes a BiEnhancer structure for low-light image enhancement that can be cascaded with a variety of HIGH-level tasks to improve performance.

**Strengths:**

Originality: This paper introduces a novel module designed to address existing challenges in the field.
Quality: The writing logic is fairly clear, and the quality is acceptable.
Clarity: The writing is overall clear.
Significance：General significance.

**Weaknesses:**

1. the proposed design cannot match the problem to be solved. This paper proposes BiEnhancer for low-light image enhancement, but from its design, there is nothing specific to low-light. In other words, it can be used for input images under any conditions.
2. lack of innovation. The main contribution of the paper is the design of the BiEnhancer module for low-light enhancement placed at the front end of multiple high-level tasks. However, the following doubts exist: firstly, as mentioned in the previous question, there is no reflection anywhere that the module is specific to low illumination. Secondly, the design of the module is not particularly attractive, and even some of the design justifications remain to be demonstrated, as shown in the following Questions for the authors.

**Questions:**

1. The innovation is insufficient. Regarding the proposed BIENHANCER, it is unclear how its design specifically caters to low-light conditions; the paper does not provide any explanation. In fact, based on the content, BIENHANCER could be understood as a generic module that, like any other module, could be integrated into any network as a component, lacking any distinctive characteristics. It could even be added to networks intended for non-low-light scenarios.
2. In section 2.1 RELATED WORK, no references are provided for the last three years.
3. For the “Ih” obtained in the proposed module, what is its role?
4. The description of Figure 2 is unclear: for the inputs of the IEM, three are given outside the blue rectangle but only two inside; how are the two outputs of the BFFM combined into one? The dimension of “Ih” is not consistent with the text.
5. The description of the proposed module in lines 173-179 does not align with the corresponding Figure 2.
6. In Figure 3, the output of (a) should be the input of (d), but the dimensions are labelled differently.
7. Figure 3(a) has only one input, but the corresponding position in Figure 2 shows two.
8. In Section 3.3, what is the function of "split"?
9. Unclear description in Section 3.4。
10. In Section 4.1, please give references for all compared methods.
11. In Section 4.2, the common method of face detection should be used instead of the method of target detection.
12. For face detection, visual results are missing.
13. Which method is “Xue et al.” in Table 4? There is no explanation or reference. In addition, why is Retinexformer used in the first two tasks and Xue et al. used in the last task?

---

### Official Review · Reviewer_9YNQ · 2024-11-01

**Soundness:** 2
**Presentation:** 2
**Contribution:** 2
**Rating:** 3
**Confidence:** 4

**Summary:**

This paper proposes a new module called BiEnhancer designed to enhance the representation of low-light images by optimizing the loss function of high-level tasks to improve performance. BiEnhancer decomposes low-light images into low-level and high-level components, adopting an attentional feature fusion strategy and a pixel-wise iterative estimation strategy to effectively enhance and restore the details and semantic information.

**Strengths:**

1.  The proposed BiEnhancer module presents a novel approach to enhancing the representation of low-light images by optimizing the loss function of high-level tasks, which is a unique perspective compared to previous methods that focus on image enhancement for human visualization.
2. The authors have conducted extensive experiments to validate the effectiveness of the BiEnhancer framework.
3. Improving the performance of high-level vision tasks in low-light conditions is a practical problem.

**Weaknesses:**

1. The innovation is not sufficient. The key issue is that the proposed architecture is similar to the previously published FeatEnHancer, including the use of similar network components.

2. The goal of constructing the BiEnhancer network module is unclear. It is not understood why the output images of the BiEnhancer network module suffered from severe quality degradation.

3. The quantitative metrics do not show significant improvement, especially in Table 4. Obvious image quality degradation can be observed in the visualized Figure 4.

4. The writing is not smooth and is difficult to understand. Particularly, the implementation details of the fusion strategy in Section 3.3 of the methodology are hard to comprehend.

**Questions:**

See the above Weaknesses.

---

### Meta-Review · Area_Chair_grfZ · 2024-12-17

**Metareview:**

All reviewers recommend rejection due to the paper’s lack of novelty, unclear design motivation, limited quantitative improvements, and weak presentation. No rebuttal was provided, and the AC found no basis to overturn the decision.

Common concerns from reviewers:

The proposed BiEnhancer shares significant architectural similarities with FeatEnhancer. The design lacks distinctiveness, with common components like skip connections, RepConv, and DWConv already widely used in the field. In addition, the rationale for BiEnhancer's design choices is not well-explained. The comparisons omit newer state-of-the-art methods. The evaluation uses only RetinaNet and Sparse R-CNN, missing modern detection frameworks.

**Additional Comments On Reviewer Discussion:**

No rebuttal was provided.

---

### Decision · Program_Chairs · 2025-01-22

Reject